# Association of Metabolomic Change and Treatment Response in Patients with Non-Alcoholic Fatty Liver Disease

**DOI:** 10.3390/biomedicines10061216

**Published:** 2022-05-24

**Authors:** Kwang Seob Lee, Yongin Cho, Hongkyung Kim, Hyunkyeong Hwang, Jin Won Cho, Yong-ho Lee, Sang-Guk Lee

**Affiliations:** 1Department of Laboratory Medicine, Yonsei University College of Medicine, Seoul 03722, Korea; kwangseob@yuhs.ac (K.S.L.); nnice2000@yuhs.ac (H.K.); hkhwang96@yuhs.ac (H.H.); 2Division of Endocrinology and Metabolism, Department of Internal Medicine, Inha University School of Medicine, Incheon 22212, Korea; choyorin@gmail.com; 3Department of Systems Biology, Glycosylation Network Research Center, Yonsei University, Seoul 03722, Korea; chojw311@yonsei.ac.kr; 4Department of Internal Medicine, Yonsei University College of Medicine, Seoul 03722, Korea; 5Institute of Endocrine Research, Yonsei University College of Medicine, Seoul 03722, Korea

**Keywords:** ezetimibe, rosuvastatin, metabolomics, non-alcoholic fatty liver disease, magnetic resonance proton-density fat fraction, magnetic resonance elastography, transient elastography, prediction

## Abstract

Non-alcoholic fatty liver disease (NAFLD) is the major cause of chronic liver disease, yet cost-effective and non-invasive diagnostic tools to monitor the severity of the disease are lacking. We aimed to investigate the metabolomic changes in NAFLD associated with therapeutic responses. It was conducted in 63 patients with NAFLD who received either ezetimibe plus rosuvastatin or rosuvastatin monotherapy. The treatment response was determined by MRI performed at baseline and week 24. The metabolites were measured at baseline and week 12. In the combination group, a relative decrease in xanthine was associated with a good response to liver fat decrease, while a relative increase in choline was associated with a good response to liver stiffness. In the monotherapy group, the relative decreases in triglyceride (TG) 20:5_36:2, TG 18:1_38:6, acetylcarnitine (C2), fatty acid (FA) 18:2, FA 18:1, and docosahexaenoic acid were associated with a decrease in liver fat, while hexosylceramide (d18:2/16:0) and hippuric acid were associated with a decrease in liver stiffness. Models using the metabolite changes showed an AUC of >0.75 in receiver operating curve analysis for predicting an improvement in liver fat and stiffness. This approach revealed the physiological impact of drugs, suggesting the mechanism underlying the development of this disease.

## 1. Introduction

Non-alcoholic fatty liver disease (NAFLD) is one of the common chronic liver diseases worldwide, and its global prevalence is increasing [1,2]. NAFLD encompasses a spectrum of liver changes, ranging from simple steatosis (non-alcoholic fatty liver) to non-alcoholic steatohepatitis (NASH) to liver cirrhosis [3]. In simple steatosis with a more benign disease course, 15–20% of patients with NASH can progress to cirrhosis over a long period of time [4,5].

NAFLD is associated with metabolic syndrome, obesity, insulin resistance, and hyperlipidemia [6] while severe liver complications are associated with comorbid type 2 diabetes mellitus [7]. Considering that the major causes of death in NAFLD patients are cardiovascular disease (CVD) and liver-associated complications, the current treatment guidelines are aimed at managing the comorbidities [3,4,8,9]. These include lifestyle modifications and pharmacological treatments such as metformin, statins, and ezetimibe [9].

Most patients with NAFLD are asymptomatic and are diagnosed by incidental findings on laboratory or imaging tests [10] Although liver biopsy is the gold standard for identifying the severity of NAFLD, it is invasive and has a high likelihood of sampling bias; thus, frequent sampling is not suitable in clinical settings [11]. To date, many non-invasive tools have been developed to evaluate fatty liver and liver stiffness, including biological markers and imaging modalities. In scoring systems, the fatty liver index (FLI) [12] or hepatic steatosis index (HSI) [13] is calculated using the body mass index (BMI) and other biochemical tests (such as aspartate aminotransferase [AST], alanine aminotransferase [ALT], and gamma-glutamyl transferase [GGT]) to determine the severity of steatosis; likewise, the FIB-4 score [14] or NAFLD fibrosis score (NFS) [15] is calculated using the platelet count, AST and ALT to determine the fibrosis stage. On the contrary, the imaging methods used include ultrasound and magnetic resonance imaging (MRI). MRI proton-density fat traction (MRI-PDFF) and controlled attenuation parameter (CAP) by transient elastography (TE) were developed to estimate the amount of liver fat. MRI-PDFF is superior to CAP [16] and correlates well with the histological grade of hepatic steatosis [17]. However, MR-based methods are not cost-effective for routine monitoring.

Beyond the traditional diagnostic tools explained above, several metabolomic studies investigating NAFLD to identify the novel biomarkers and unveil the pathophysiology of NAFLD have been published. Branched-chain amino acids have been associated with NASH but not with steatosis, and their levels were decreased in the advanced fibrotic stage [18,19]. Lipidomic studies have shown that plasma monounsaturated fatty acids are associated with NAFLD, and triglyceride (TG) species with a low carbon number and double-bond content are increased in patients with NAFLD [20,21]. Many studies have revealed the different metabolomic patterns between healthy individuals and NAFLD patients; however, conventional methods cannot determine the metabolomic patterns and changes that could monitor the efficacy of the treatment.

Hence, this study aimed to identify the metabolites associated with the response to treatment in NAFLD patients. We hypothesised that the metabolite change from the baseline concentration indicates the metabolic alterations caused by the drug. To investigate this, a metabolomic study was performed before and after administering two types of pharmacological treatment (ezetimibe + rosuvastatin vs. rosuvastatin).

## 2. Materials and Methods

### 2.1. Patient Selection

Detailed information on the clinical trial protocol has been described previously [22]. Participants who met the following inclusion criteria were enrolled in the study: (1) aged 19–80 years; (2) with hepatic steatosis detected on abdominal ultrasound; (3) with hyperlipidaemia who had one or less risk factor (smoking, hypertension or hypertensive medication, low levels of high-density lipoprotein cholesterol (HDL-C < 40 mg/dL), aged ≥ 45 years for men or ≥55 years for women, early coronary artery disease in first-degree relatives (aged < 55 years for men and <65 years for women), and high level of HDL-C (>60 mg/dL) redeems one risk factor) + low-density lipoprotein cholesterol (LDL-C) level of ≥130 mg/dL or two risk factors + LDL-C level of ≥100 mg/dL or high risk (>50% risk of developing carotid stenosis, abdominal aneurysm, and diabetes) + LDL-C level of ≥70 mg/dL; (4) evidence of atherosclerosis on carotid ultrasound. By contrast, patients who (1) had type 1 diabetes or secondary diabetes; (2) were treated with ezetimibe or had a history of discontinuing ezetimibe/statin treatment due to the occurrence of adverse events; (3) had uncontrolled diabetes (change of hypoglycaemic agents within the 12-week period prior to screening or an HbA1C level of >9.0%); (4) had acute or chronic metabolic acidosis or ketoacidosis within 6 months prior to screening; (5) received thiazolidinedione or sodium-glucose cotransporter 2 inhibitors, or amiodarone, methotrexate tamoxifen, valproate, and corticosteroids, which could reduce or induce fatty liver; (6) had alcohol intakes of >210 g/week for men and 140 g/week for women or were positive for hepatitis (A, B, or C), haemochromatosis, Wilson’s disease, liver cancer, autoimmune hepatitis, or liver cirrhosis (Child-Pugh score of >7, platelet count of <75,000/mm^2^, prothrombin time of >16 s); (7) had a history of corticosteroid treatment for at least 14 days within 8 weeks prior to screening; (8) received cancer treatment, including chemotherapy or radiotherapy within 2 years; (9) had galactose/lactose intolerance or genetic issues in glucose-galactose absorption; (10) had malnutrition, general weakness, hypopituitarism, adrenal failure, or abnormal liver function (ALT, AST, alkaline phosphatase (ALP), or total bilirubin > upper normal value × 5); (11) received weight-loss medication or experienced drug intoxication within 12 weeks prior to screening; (12) were infected with human immunodeficiency virus; had a history of renal dysfunction (with estimated glomerular filtration rate of <45 mL/min/1.73 m^2^ or undergoing dialysis); (13) had haemoglobin levels of <10.5 g/dL, heart failure (New York Heart Association class III and IV), uncontrolled heart arrhythmia, or cardio-cerebral events (unstable angina, myocardial infarction, transient ischaemic attack, cerebral infarct, or haemorrhage) within the 12 weeks prior to screening; (14) were pregnant or nursing mothers; (15) had medical conditions that could affect the drug metabolism (e.g., history of gastrointestinal tract surgery, active inflammatory bowel disease within 1 year) were excluded.

### 2.2. Study Design

An automated allocation sequence was used to assign the participants to two treatment groups at a 1:1 ratio. The participants were stratified according to the status of type 2 diabetes before randomisation to equalise the proportion of patients with type 2 diabetes in both groups. The patients were enrolled in a 24-week, randomised, open-label, prospective trial of oral medication using rosuvastatin (5 mg) + ezetimibe (10 mg) (E/S group) administered once daily or rosuvastatin (5 mg) alone (S group). Participants with a history of statin treatment had at least two weeks of drug wash-out period prior to the baseline investigation. Anthropometric measurements and medical history were noted during the first interview. During the study, the patients were asked to visit the clinic for initial screening and measurements at weeks 12 and 24 (Appendix A). Informed consent was obtained to use the remnant specimen for further analysis.

### 2.3. Laboratory and Image Testing

BMI, waist circumference, and fasting blood test samples were collected at baseline and week 12 visits. The laboratory values included complete blood count, liver enzymes (AST, ALT, and GGT), total bilirubin, creatinine, blood urea nitrogen, uric acid, total protein, albumin, ALP, C-reactive protein, ferritin, fasting plasma glucose, haemoglobin A1c (HbA1c), insulin, and lipid profile (total cholesterol, LDL-C, HDL-C, and TG). At the baseline and the last visit (week 24), CAP and liver stiffness measurement (LSM) were obtained using TE (FibroScan^®^; Echosens, Paris, France). MRI-PDFF and MR elastography (MRE) were performed on the same date of visit.

### 2.4. Metabolite Measurements

Metabolite measurements were performed at baseline and week 12. Metabolite quantification of 630 metabolites, including lipids, was performed in plasma using a tandem mass spectrometry MxP^®^ Quant 500 Kit (Biocrates Life Sciences, Innsbruck, Austria). The measurement data quality was evaluated based on the following conditions: (1) coefficient of variance for reference standards < 25%; (2) 50% of the metabolites in the reference > limit of detection; (3) 50% of the metabolites in samples > limit of detection. As a result, 436 metabolites qualified for further analysis. Additionally, short-chain fatty acids (acetic acid, propionic acid, and butyric acid) and metabolites involved in the tricyclic acid cycle (citrate, cis-aconitate, isocitrate, α-ketoglutarate, succinate, fumarate, malate, lactate, pyruvate, and oxaloacetate) were also quantified by conducting laboratory-developed tests using liquid chromatography-tandem mass spectrometry. If the concentration of a metabolite in a subject was zero, the concentration was considered to be half of the smallest concentration among participants with non-zero values. The concentration was expressed in μM.

### 2.5. Main Outcome

The main outcome of the patient was a change in the MRI-PDFF and MRE. A good response to the changes in liver fat was defined as a ≥ 30% decrease in the MRI-PDFF values relative to the baseline, while a good response to liver stiffness was defined as a decrease in MRE relative to the baseline. Another outcome of the patient was the change in CAP and LSM measured using TE. With TE, a good response to liver fat was defined as the downgrade of steatosis classified based on the CAP values (238 dB/m for S ≥ 1, 260 dB/m for S ≥ 2, and 293 dB/m for S ≥ 3) [23]. A good response to liver stiffness was defined as a decrease in LSM value compared with the baseline values.

### 2.6. Statistical Analysis

All continuous variables were expressed as medians and interquartile ranges. Wilcoxon’s rank-sum test was used to evaluate the differences between groups. A *p* value of <0.05 was considered significant. Multiple comparisons were corrected using the false discovery rate (FDR) method (FDR-adjusted *p* value < 0.1). The percent change in a metabolite was calculated as the absolute concentration change divided by the baseline concentration × 100. The fold change was calculated as the log_2_ transformation of the ratio between the post-treatment and pre-treatment median metabolite abundance. Fold-change analysis was performed by plotting the log transformation of the *p* value obtained using paired Wilcoxon’s test and the calculated fold change for each metabolite. Metabolites with a fold change higher than 2, and a *p* value of less than 0.05 were considered significant. Linear regression was used to observe the relationship between the imaging test outcomes and metabolites. The diagnostic performance of an individual metabolite in classifying good and non-responders was analysed using the receiver operating characteristic (ROC) curve and its area under the curve (AUC). Statistical analysis and visualisation were performed using the R software (version 4.0.4).

### 2.7. Ethics

The study was approved by the institutional review board of Yonsei University College of Medicine (IRB no. 4-2020-0843).

## 3. Results

### 3.1. Response in Clinical Parameters and Metabolites

Metabolomic analyses were conducted in 64 participants, including 31 patients in the E/S group and 32 patients in the S group. One patient in the S group was excluded because the measurement of the main outcome was performed at week 36. Irrespective of the treatment response, a fold-change analysis was performed to observe the metabolomic profile changes before and after treatment (Figure 1). In the E/S group, the TG 18:2_28:0, TG 22:6_32:0, TG 16:0_30:2, TG 22:5_32:0, TG 18:2_33:0, TG 14:0_36:4, TG 16:0_36:5, butyric acid, HexCer (d16:1/24:0), and propionic acid levels were significantly altered after treatment. In the S group, the levels of several TG species, including TG 18:2_28:0, TG 16:0_30:2, TG 22:5_32:0, the sum of saturated and monounsaturated TGs, CE (14:1), butyric acid, cystine, and propionic acid, were significantly changed after treatment. Propionic acid was the only metabolite that increased after treatment in both study groups. When treatment response was defined on the basis of the MRI-PDFF (≥30% relative decrease), 52% (16/31) of patients in the E/S group and 44% (14/32) of patients in the S group showed a good treatment response to liver fat, although the proportion difference was not significant. When the good and non-response patients in the E/S group were compared, the baseline concentration of acetic acid was significantly lower in the good responders, whereas the good responders in the S group showed significantly higher levels of free fatty acids (Table 1). The absolute values of baseline MRI-PDFF or CAP were not significantly different between the good and non-responders.

### 3.2. Association of the Treatment Response Defined by MRI-PDFF or MRE Results and Metabolomic Changes

Owing to the inter-individual variability of the metabolites, the percent change in the concentration of each metabolite was compared to the baseline concentration. In the E/S group, the xanthine level was significantly altered in good responders, whereas the acetylcarnitine (C2), octadecenoic acid (FA 18:1), octadecadienoic acid (FA 18:2), docosahexaenoic acid (DHA), TG 20:5_36:2, and TG 18:1_38:6 levels were significantly altered in good responders in the S group. The levels of these metabolites tended to decrease in good responders (Table 2a and Appendix A). When a good response was determined to be a decrease in the frequency of MRE, 14 patients in the E/S group and 16 patients in the S group were found to show a good response; the levels of the following metabolites were significantly increased in the good responders: choline in the E/S group; HexCer (d18:2/16:0), and hippuric acid in the S group (Table 2b and Appendix A). Linear regression analysis of the percent changes in these metabolites with the outcome values showed significant linear relationships (Figure 2). To ascertain whether these metabolites could determine the status of treatment response, an ROC analysis was performed, and the AUC was calculated. In predicting steatosis improvement, the percent change in xanthine showed an AUC of 0.763 in the E/S group, and the combination of six metabolites (C2, FA 18:2, FA 18:1, DHA, TG 20:5_36:2, and TG 18:1_38:6) showed an AUC of 0.919 in the S group. In predicting liver stiffness improvement measured by MRE, the percent change of choline showed an AUC of 0.824 in the E/S group, while the combination of two metabolites (HexCer (d18:2/16:0) and hippuric acid) showed an AUC of 0.875 in the S group (Figure 3). This finding suggests that changes in the levels of metabolites relative to the baseline levels could predict the dynamic metabolomic status and responsiveness of the patients to the treatment (Appendix A).

### 3.3. Association of the Treatment Response Defined by TE and Metabolomic Changes

Additionally, a replication analysis was performed between the good and non-responders determined by the CAP and LSM values measured by the TE. Because TE is known to have inferior diagnostic performance compared with MRI, only patients with categorical agreement of therapeutic response were included in the comparison between MRI and TE. The percent changes in xanthine levels and many TG species in the E/S group and percent changes in the C2, FA 18:2, FA 18:1, DHA, DG 16:1_18:1, and DG 16:1_18:2 levels in the S group were significantly different between the good and non-responders to liver fat. In addition, the percent change in choline levels in the E/S group and percent change in hippuric acid and hypoxanthine levels in the S group were significantly different between the good and non-responders in terms of liver stiffness. Consequently, the effect of xanthine, C2, FA 18:2, FA 18:1, and DHA in liver fat and that of choline and hippuric acid in liver stiffness were found to be significant (Appendix A).

## 4. Discussion

Despite the increasing proportion of the disease population worldwide, the treatment strategies for NAFLD are not well established, and more diagnostic modalities should be developed. The pathophysiology of hepatic steatosis and its progression to NAFLD and NASH is largely unknown; however, excessive triglyceride production and secretion due to fatty acid uptake and hepatic de novo lipogenesis cause the accumulation of lipids in the liver [24,25]. By contrast, a high-cholesterol diet was also reported to induce NAFLD [26].

In this study, the metabolomic profile of patients treated with a combination of ezetimibe and rosuvastatin was compared with that of patients treated with rosuvastatin alone. Ezetimibe, an inhibitor of Niemann-Pick C1-like 1 protein located at the brush border of the intestine, was shown to improve hepatic steatosis, but the benefits in steatohepatitis or cirrhosis remain unclear [27,28]. Statins are recommended in patients with NAFLD to reduce the risk of CVD [9]. Furthermore, treatment with a combination of ezetimibe and statins was reported to be beneficial in patients at a high risk of cardiovascular events [29]. Our study revealed that the absolute value of MRI-PDFF significantly decreased after treatment in the E/S group, while that in the S group did not show significance.

The fold-change analysis showed changes in the levels of several metabolites after treatment with drugs, irrespective of the response to NAFLD treatment. The butyric acid levels were significantly decreased, while the propionic acid levels were significantly increased after treatment in both groups. Statin treatment was thought to induce this change since the two groups commonly used statins. The butyric acid levels decreased in statin-treated patients owing to the changes in the gut microbiome [30]; however, evidence to support the increase in propionic acid levels in statin-treated patients is yet to be established. Propionic acid inhibits cholesterol and triacylglycerol synthesis in human hepatocytes [31] but considering the efficacy of statins in preventing CVD rather than NAFLD, propionic acid treatment may be associated with the improvement in cardiovascular metabolism. Basic research has shown that it attenuates atherosclerosis by regulating the immune system of the gut [32] and protects against cardiovascular damage by activating the regulatory T-cell-dependent pathways [33]. These trends of changes in short-chain fatty acids by statin therapy remain controversial across studies [34,35], and further studies are needed to confirm the impact of these metabolites. In addition, cystine levels and the sum of saturated/monounsaturated TGs were significantly altered in the S group. Cystine depletion in the S group can be explained by activation of the skeletal muscle system xC− (cystine/glutamate antiporter) induced by statin therapy, which causes myalgia [36].

More importantly, in the E/S group, an association was observed between decreased xanthine level and improvement of liver fat and between increased choline level and improvement of liver stiffness (Figure 4). Xanthine is an intermediate metabolite involved in purine catabolism. Xanthine oxidoreductase (XOR) converts hypoxanthine to xanthine and xanthine to uric acid and reactive oxygen species (ROS), which activate the NLR family pyrin domain containing 3 (NLRP3) inflammasome which is associated with the development of NAFLD [37]. Higher hypoxanthine levels were associated with obesity and were secreted from adipose tissues under hypoxic conditions [38,39]. Increase in plasma XOR activity was induced by the disease [40]. The decrease in xanthine levels with the improvement in liver fat can be explained by the fact that ezetimibe attenuates steatohepatitis through the induction of autophagy and the inhibition of NLRP3 inflammasome [41] which would result in the downregulation of the purine catabolism pathway. Ezetimibe may interrupt the vicious cycle of purine catabolism activation by activating XOR and NAFLD induced by the NLRP3 inflammasome.

Further research is required to validate whether change in choline abundance is associated with improvement in liver stiffness; however, methionine- and choline-deficient diets induce NASH in mouse models [42]. Choline is the major source of phosphatidylcholine required to assemble VLDL, which plays a role in TG secretion from the liver [43]. The accumulation of TGs increases ROS production, mitochondrial DNA damage, and apoptosis [44]. Several possible mechanisms can explain the effect of ezetimibe on the choline concentration. First, ezetimibe could modify the gut microbiota population, which metabolises choline to trimethylamine and interrupts the absorption of dietary choline [45,46]. Second, since the NLRP3 inflammasome is associated with inflammatory bowel disease [47], NLRP3 inflammasome inhibition by ezetimibe treatment could attenuate inflammation in the gut, stabilising the gut microbiome that helps absorb choline. Since plasma choline level is not a reliable marker for choline intake or absorption, our explanation remains to be debated [48,49]. On the other hand, ezetimibe treatment seems to restrict the availability of cholesterol in the liver, thus hindering cholesterol-poor lipoprotein synthesis [50]. Choline is converted to phosphatidylcholine in the liver to produce VLDL, while ezetimibe blocks the synthesis of VLDL [51,52], diminishing the use of the choline pool. Consequently, choline, as a mediator of healthy conditions, decreases metabolic stress and may prevent liver damage [43,53].

In the S group, the C2, FA 18:2, FA 18:1, and DHA levels decreased with the reduction in the amount of liver fat. We expect that the reduction in the levels of these metabolites is a consequence of rosuvastatin treatment rather than the underlying mechanisms of NAFLD because these lipid metabolites are the sources of hepatic lipid accumulation. Statins inhibit the production of cholesterol, which in turn significantly decreases the level of saturated and monounsaturated FAs in total FAs, and exceptionally, octadecenoic acid (C18:1) and octadecadienoic acid (C18:2) [54]. Another study showed that strong statins, such as rosuvastatin, decreased the DHA levels, although the mechanism is unknown, suggesting that DHA added to the statin would improve the clinical effects [55].

The hippuric acid was upregulated with an increase in liver stiffness. Hippurate is reported to reflect hepatic function because it is produced from the metabolism of benzoate, which is mainly stored in the liver mitochondria [56] and is an indicator of functional hepatic reserve [57]. Many studies have also shown that elevated hippurate levels are associated with improvement in hepatic steatosis and good metabolic health [58,59,60]. Notably, higher hippurate levels were associated with gut microbiome diversity [59] whereas rosuvastatin had a beneficial impact on gut microbiome diversity [35,61]. Rosuvastatin treatment changed the gut microbiome in our patients, and the increase in hippurate absorption induced the recovery of the liver tissue. The causal relationship between hippurate and metabolic disease has not been studied thoroughly. However, with an unknown mechanism, the hippurate level indicates the status of hepatic function or is a key mediator in the improvement of liver damage.

We have also performed comparisons of baseline metabolite levels between the responders and non-responders classified by MRI-PDFF (Appendix A). In the E/S group, Hex3Cer (d18:1/16:0) was significantly lower in the responders. In the S group, ornithine, methionine sulfoxide, homocysteine, and kynurenine were significantly lower and FA 18:2, FA 18:1, DG 18:2_18:2, and DG 16:1_18:2 were significantly higher in the responders. However, the evidence of these metabolites’ association with NAFLD is insufficient to interpret and needs further research.

Our study has some limitations. Imaging studies were performed at week 24, and biochemical tests were performed at week 12. Caution is needed when interpreting the results, which may underestimate the change in imaging findings. Second, treatment with a combination of statin and ezetimibe hindered the exact effect of ezetimibe treatment on the metabolomic profiles.

The percent change in metabolite levels was well correlated with the percent change in MRI-PDFF and MRE values. Due to the inter- and intra-individual variance of metabolites in the real world, examining the changes in metabolite levels is another feasible option for the investigation of diseases. Notably, we have observed different metabolomic changes with different medications used although the outcome (liver fat reduction) was identical. It provides a clue for the reason of various metabolite associations observed by different studies in the same disease. Researchers are required to stratify patients in detail to discover novel biomarkers. Furthermore, studies on these metabolites along with their treatments can provide additional clues for developing a novel combination treatment, simply from supplementation of a significant metabolite to other metabolites involved in the drug’s mechanism of action.

## 5. Conclusions

In conclusion, different metabolite patterns were observed in different treatment strategies. Response to treatment was associated with metabolite pattern changes during the treatment. These data demonstrate the putative mechanism of the drugs and the disease.

## Figures and Tables

**Figure 1 biomedicines-10-01216-f001:**
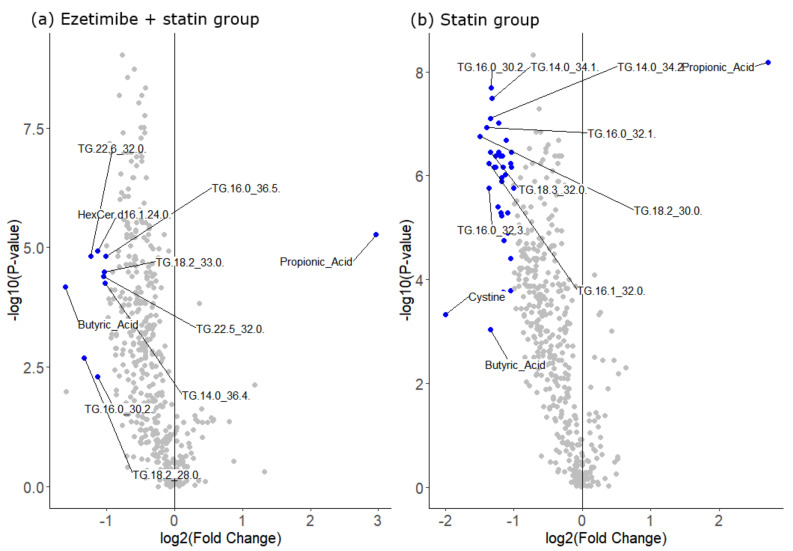
Fold-change analysis of metabolite concentrations between pre- and post-treatment by treatment groups, (**a**) ezetimibe + statin group and **b** statin monotherapy group. Significance of the fold change was defined by an FDR of < 0.05 and a fold change value of ≥2 (metabolites indicated by blue dots). In (**b**), triglyceride species with log_2_ (fold change) > −1.3 are not labelled.

**Figure 2 biomedicines-10-01216-f002:**
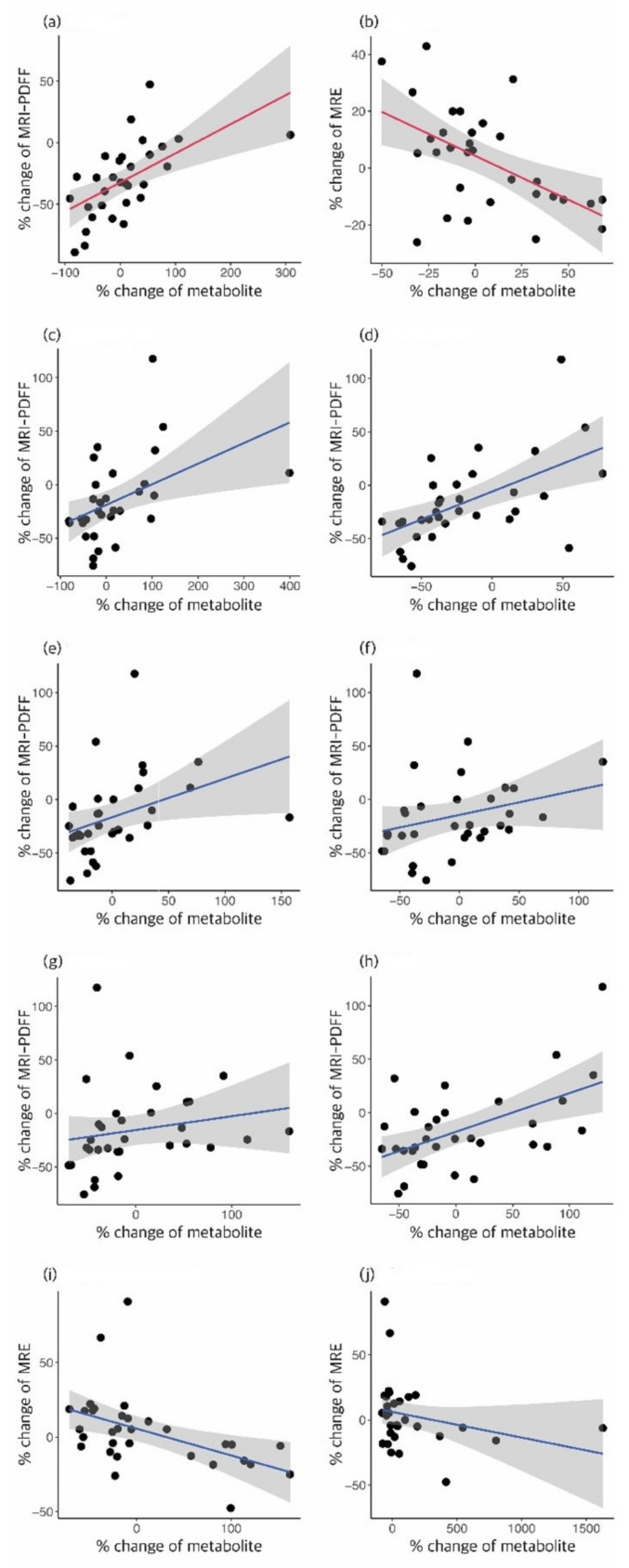
Linear regression analysis of the associated percent changes of the metabolite levels and MRI diagnostic values. Linear correlation of (**a**) xanthine with MRI-PDFF change and (**b**) choline with MRE change in ezetimibe + statin group. Linear correlation of (**c**) TG 20:5_36:2, (**d**) TG 18:1_38:6, (**e**) C2, (**f**) FA 18:2, (**g**) FA 18:1, and (**h**) DHA with MRI-PDFF change, and (**i**) HexCer (d18:2/16:0) and (**j**) hippuric acid with MRE change in the statin group. The shaded area corresponds to the 95% confidence intervals.

**Figure 3 biomedicines-10-01216-f003:**
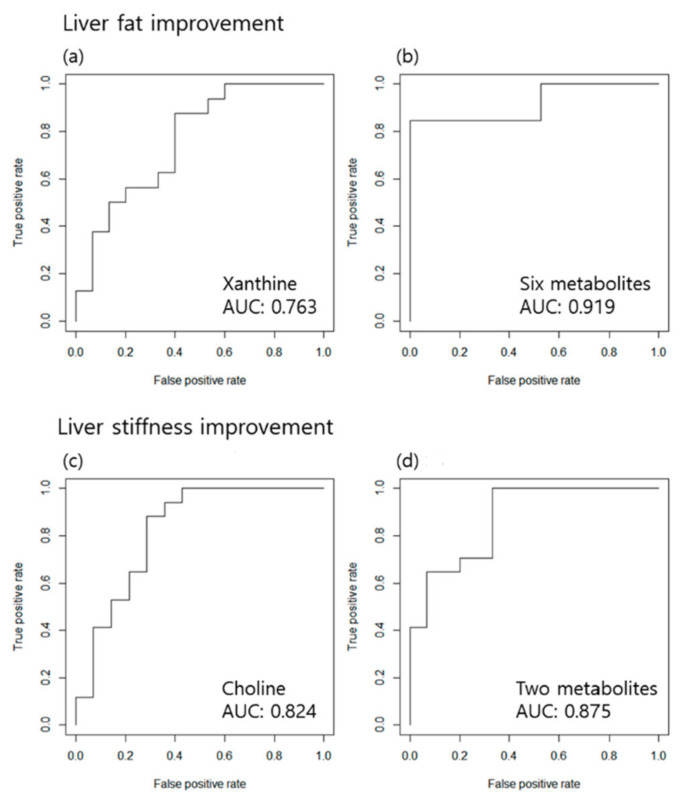
ROC analysis and AUC values for classifying good responders using the relative change in metabolite levels. Prediction of liver fat improvement measured by MRI-PDFF in the (**a**) ezetimibe + statin group and (**b**) statin group. Prediction of liver stiffness improvement measured by MRE in the (**c**) ezetimibe + statin group and (**d**) statin group.

**Figure 4 biomedicines-10-01216-f004:**
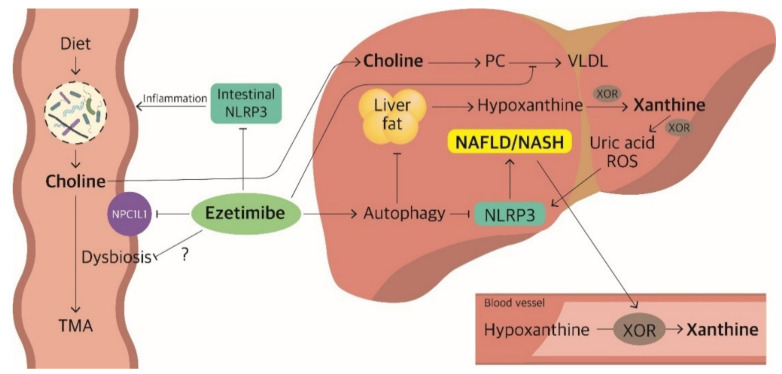
Putative schematic model of the relationships between the metabolites and ezetimibe. PC, phosphatidylcholine; VLDL, very-low-density lipoprotein; XOR, xanthine oxidoreductase; TMA, trimethylamine; NPC1L1, Niemann-Pick C1-Like 1; ROS, reactive oxygen species; NLRP3, NLR family pyrin domain containing 3.

**Table 1 biomedicines-10-01216-t001:** Baseline demographic data and characteristics.

Treatment	Ezetimibe + Statin Group (*n* = 31)			Statin Group (*n* = 32)		
Response in liver fat *	Good response (*n* = 16)	Non-response (*n* = 15)	*p* value	Good response (*n* = 14)	Non-response (*n* = 18)	*p* value
Age, years	46.7 [43.5–55.0]	49.4 [43.8–55.6]	0.68	52.5 [45.3–56.8]	59.4 [42.2–68.2]	0.16
Sex			0.30			>0.99
Male (%)	9 (56%)	12 (80%)		7 (50%)	10 (56%)	
Female (%)	7 (44%)	3 (20%)		7 (50%)	8 (44%)	
BMI, kg/m^2^	28.9 [25.0–31.7]	29.5 [24.9–30.8]	0.98	27.6 [26.7–28.7]	28.4 [27.8–31.0]	0.11
HTN			0.36			>0.99
No (%)	7 (44%)	10 (67%)		9 (64%)	12 (67%)	
Yes (%)	9 (56%)	5 (33%)		5 (36%)	6 (33%)	
History of DM			0.18			>0.99
No (%)	2 (13%)	6 (40%)		4 (29%)	5 (28%)	
Yes (%)	14 (88%)	9 (60%)		10 (71%)	13 (72%)	
Image findings						
MRI-PDFF, %	17.9 [12.5–23.6]	15.4 [12.9–20.1]	0.45	13.1 [11.2–16.0]	12.4 [10.4–17.9]	>0.99
CAP, dB/m	327.5 [315.5–335.0]	327.0 [287.0–333.0]	0.31	305.0 [284.0–325.0]	324.5 [299.0–333.0]	0.14
MR elastography, kPa	2.1 [1.8–2.5]	1.8 [1.6–2.4]	0.14	1.9 [1.7–2.7]	2.2 [1.9–2.4]	0.28
LSM, kPa	6.3 [5.4–7.4]	5.7 [4.4–8.4]	0.62	6.2 [5.1–6.8]	6.8 [4.9–8.7]	0.23
Lipid profile						
Free fatty acid, mmol/L	472.0 [397.5–569.0]	575.0 [429.5–724.5]	0.29	620.0 [550.0–750.0]	426.0 [334.0–515.0]	0.004
Acetic acid, μM	69.0 [51.4–91.9]	93.4 [66.7–159.2]	0.03	97.6 [67.8–162.7]	68.2 [61.0–101.9]	0.20
Butyric acid, μM	2.7 [2.2–3.9]	2.8 [0.8–3.5]	0.84	2.5 [1.0–3.4]	2.8 [0.8–3.6]	0.99
Propionic acid, μM	0.6 [0.6–2.4]	0.9 [0.6–2.8]	0.62	0.8 [0.6–2.2]	1.8 [0.7–3.8]	0.27
Total cholesterol, mg/dL	189.5 [162.0–204.5]	218.0 [187.0–244.5]	0.06	209.0 [186.0–254.0]	196.5 [188.0–214.0]	0.46
HDL cholesterol, mg/dL	42.5 [36.0–47.5]	49.0 [40.5–54.5]	0.10	43.5 [40.0–49.0]	42.0 [39.0–46.0]	0.61
LDL cholesterol, mg/dL	104.8 [81.1–121.2]	126.4 [102.0–162.0]	0.14	114.3 [86.4–160.2]	108.5 [101.4–139.6]	0.99
Triglycerides, mg/dL	175.0 [128.5–257.5]	158.0 [125.0–183.0]	0.49	272.5 [218.0–309.0]	210.5 [146.0–273.0]	0.10

* Good response to the changes in liver fat was defined as ≥30% decrease in MRI-PDFF values relative to the baseline values.

**Table 2 biomedicines-10-01216-t002:** Significantly different changes in metabolite levels between the good and non-response groups after receiving the two treatments (ezetimibe + statin and statin alone). (**a**) Significant metabolites associated with the improvement in MRI-PDFF values. (**b**) Significant metabolites associated with the improvement of MR elastography value. Metabolite changes were defined as the change in concentration compared with the baseline values. A *p* value < 0.05 was considered significant, and the FDR *p* value < 0.1.

(a) Steatosis	Metabolite	% Change from the Baseline [Q1-Q3]		*p* Value	FDR *p* Value
		Good Responder	Non-Responder		
Ezetimibe + statin	Xanthine	−21.4 [−59.1 to +10.4]	+19.5 [−7.3 to +64.9]	0.01	0.02
Statin	TG 20:5_36:2	−43.5 [−51.1 to −26.5]	+14.2 [−15.0 to +92.3]	<0.001	0.04
	TG 18:1_38:6	−53.2 [−63.2 to −42.4]	−13.8 [−37.2 to +23.3]	<0.001	0.04
	C2	−22.2 [−30.0 to −17.1]	+5.4 [−12.3 to +29.5]	<0.001	0.007
	FA 18:2	−39.0 [−60.3 to −6.3]	+8.6 [−18.1 to +40.0]	0.001	0.009
	FA 18:1	−42.9 [−51.4 to −19.1]	+15.8 [−28.1 to +52.3]	0.003	0.009
	DHA	−36.1 [−46.0 to −17.4]	+13.3 [−20.5 to +78.2]	0.009	0.02
**(b) Stiffness**	**Metabolite**	**% Change from the Baseline [Q1-Q3]**		***p* ** **Value**	**FDR *p* Value**
		**Good Responder**	**Non-Responder**		
Ezetimibe + statin	Xanthine	−21.4 [−59.1 to +10.4]	+19.5 [−7.3 to +64.9]	0.01	0.02
Statin	HexCer(d18:2/16:0)	+81.0 [−21.7 to +107.4]	−25.6 [−49.1 to −9.3]	0.003	0.07
	Hippuric acid	+40.2 [−11.0 to +391.1]	−28.0 [−48.0 to +12.5]	0.006	0.02

## Data Availability

All the data presented in this study are available from the corresponding author upon reasonable request.

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
