# Peer review of "Association of Metabolomic Change and Treatment Response in Patients with Non-Alcoholic Fatty Liver Disease"

_biomedicines, 2022, doi:10.3390/biomedicines10061216_

Round 1
Reviewer 1 Report
This paper examines the metabolomic changes in response to two different drug treatments for NAFLD. Their results identified a few metabolites that changed differentially in responders versus non-responders of different treatments. While the results are intriguing and could potentially enlighten new biomarker development, here are some questions to consider:
- Line 81. Did the inclusion criteria account for the different severity of NAFLD? Was that accounted for in the analysis?
- line 139: were the metabolites also measured in fasting serum samples?
- Methods: Was there any dietary intake data available during the intervention as this may affect metabolite status.
- Results: (1)Are there any metabolites altered in both treatment groups, i.e. , would it be possible to combine the responders in the two groups to compare them with the non-responders in the two groups to identify metabolites that are more “universal” to different treatments?
(2) why are the metabolite changes treatment-specific? How will they affect the implication of the study?
(3) Are there any metabolites at baseline that can predict responder or not, because this is the most urgent need for biomarker identification and potentially treatment plan decision?
- Discussion: Line 325. It should be noted that choline level in the blood is not a reliable marker of choline absorption or dietary intake, especially during metabolic stress.
Reviewer 2 Report
Biomedicines
General comment- This is a clearly presented and well-written paper. NAFLD affects 25% of the Western world population; however, the mechanisms that drive the transition from NAFLD to NASH or Cirrhosis are still unclear. In this study, the authors investigated the metabolomic changes in NAFLD patients associated with therapeutic responses. Ezetimibe plus rosuvastatin and rosuvastatin monotherapy was used for 24-wks in NAFLD patients.
Summary of the salient findings:
In combination therapy, the relative decrease in xanthine was associated with a reduction in liver fat, and relative increase in choline was associated with a response to liver stiffness.
Whereas, in monotherapy group, the relative decreases in TG, acetyl carnitine (C2), fatty acids (FA), and docosahexaenoic acid were associated with a decrease in liver fat. In contrast, hippuric acid was associated with decrease in liver stiffness.
The proposed study is interesting, but I have the following comments and concerns.
- The study delineate the changes in metabolomic profile in NAFLD in response to therapy. However, it’s not clear whether these drugs alters the liver function (e.g. ALT, ALP etc.) and lipid profile (such as VLDL, LDL) in these patients. Any explanation.
- Also, did you employ some NAFLD patients as control/placebo without any therapy. If yes, how was the metabolomic profile change in drug-treated one as compared to placebo/control.
- 1. a) Some of the significant metabolites are not labeled, it will be great if you can label them. In panel (b), label the ones with very significant neg p-value and fold change.
I recommend the manuscript be accepted for publication, with addressing these concerns
